# Prognostic Value of Baseline ^18^F-FDG PET/CT to Predict Brain Metastasis Development in Melanoma Patients

**DOI:** 10.3390/cancers16010127

**Published:** 2023-12-26

**Authors:** Forough Kalantari, Seyed Ali Mirshahvalad, Magdalena Hoellwerth, Gregor Schweighofer-Zwink, Ursula Huber-Schönauer, Wolfgang Hitzl, Gundula Rendl, Peter Koelblinger, Christian Pirich, Mohsen Beheshti

**Affiliations:** 1Division of Molecular Imaging and Theranostics, Department of Nuclear Medicine, University Hospital Salzburg, Paracelsus Medical University, 5020 Salzburg, Austria; foroughkalantari@gmail.com (F.K.); mirshahvalad.sa@gmail.com (S.A.M.); g.schweighofer-zwink@salk.at (G.S.-Z.); u.huber-schoenauer@salk.at (U.H.-S.); g.rendl@salk.at (G.R.); c.pirich@salk.at (C.P.); 2Department of Nuclear Medicine, University Hospital, Iran University of Medical Sciences, 1461884513 Tehran, Iran; 3Joint Department of Medical Imaging (University Medical Imaging Toronto (UMIT)), University Health Network, Mount Sinai Hospital–Women’s College Hospital, University of Toronto, Toronto, ON M5G 2N2, Canada; 4Department of Dermatology, University Hospital Salzburg, Paracelsus Medical University, 5020 Salzburg, Austria; m.hoellwerth@salk.at (M.H.); p.koelblinger@salk.at (P.K.); 5Biostatistics and Publication of Clinical Trial Studies, Research and Innovation Management (RIM), Paracelsus Medical University, 5020 Salzburg, Austria; w.hitzl@salk.at; 6Department of Ophthalmology and Optometry, Paracelsus Medical University, 5020 Salzburg, Austria; 7Research Program Experimental Ophthalmology & Glaucoma Research, Paracelsus Medical University, 5020 Salzburg, Austria

**Keywords:** melanoma, FDG PET/CT, survival, brain metastasis, predictive value

## Abstract

**Simple Summary:**

Malignant melanoma is an aggressive malignancy with an unproportionally high mortality rate among skin-related malignancies, being the third most common cancer that metastasizes to the brain. Approximately 10% of patients with a history of invasive melanoma ultimately develop brain metastases. In metastatic patients, this presentation rate increases up to 60%. ^18^F-FDG PET/CT is an accurate imaging modality for detecting melanoma metastases. However, the well-known limitation of ^18^F-FDG PET/CT is its unfavourable tumour-to-background uptake ratio in the brain. Thus, as the mainstay assessment tool for distant metastasis, the prognostic value of ^18^F-FDG PET/CT-derived parameters is important to determine, especially considering the brain. However, regarding prognostication, the majority of previous studies have included small populations of advanced-stage melanoma patients. Also, data regarding the baseline prognostic value of ^18^F-FDG PET/CT in melanoma patients with brain metastases are scarce. Thus, we aimed to investigate its prognostic value in predicting the occurrence of brain metastases in melanoma patients.

**Abstract:**

To investigate the value of ^18^F-FDG-PET/CT in predicting the occurrence of brain metastases in melanoma patients, in this retrospective study 201 consecutive patients with pathology-proven melanoma, between 2008 and 2021, were reviewed. Those who underwent ^18^F-FDG-PET/CT for initial staging were considered eligible. Baseline assessment included histopathology, ^18^F-FDG-PET/CT, and brain MRI. Also, all patients had serial follow-ups for diagnosing brain metastasis development. Baseline ^18^F-FDG-PET/CT parameters were analysed using competing risk regression models to analyze their correlation with the occurrence of brain metastases. Overall, 159 patients entered the study. The median follow-up was six years. Among clinical variables, the initial M-stage and TNM-stage were significantly correlated with brain metastasis. Regarding ^18^F-FDG-PET/CT parameters, regional metastatic lymph node uptake values, as well as prominent SULmax (pSULmax) and prominent SUVmean (pSUVmean), were significantly correlated with the outcome. Cumulative incidences were 10% (6.3–16%), 31% (24.4–38.9%), and 35.2% (28.5–43.5%) after 1, 5, and 10 years. There were significant correlations between pSULmax (*p*-value < 0.001) and pSULpeak (*p*-value < 0.001) and the occurrence of brain metastases. The higher these values, the sooner the patient developed brain metastases. Thus, baseline ^18^F-FDG-PET/CT may have the potential to predict brain metastasis in melanoma patients. Those with high total metabolic activity should undergo follow-up/complementary evaluations, such as brain MRI.

## 1. Introduction 

Skin cancer is the most commonly diagnosed cancer type worldwide [1]. Malignant melanoma, an aggressive, highly metastatic malignancy, has the highest mortality rate among all skin-related malignancies, imposing a healthcare burden and cost, particularly in advanced stages [2,3,4,5]. Thus, it is a challenge in global public health and has been shown to have a continuous increase in incidence during the last 50 years, particularly in fair-skinned populations [1,4].

Moreover, melanoma is the third most common metastatic tumour to the brain (after lung and breast cancer), accounting for roughly 10% of all developed brain metastases [6,7]. It is noteworthy that, regardless of the disease stage, approximately 10% of patients with a history of invasive melanoma ultimately develop brain metastases [8,9]. In metastatic patients, this presentation rate increases up to 60% [7] and, in post-mortem autopsy, it has been shown to affect over 70% of stage IV patients [10]. Also noteworthy is the fact that asymptomatic brain metastases were reported in approximately 12% of patients with metastatic melanoma [11].

Some clinical predictors are known as risk factors for brain metastasis development in melanoma patients, such as young age, male sex, ulceration, depth of invasion (Breslow thickness), and locoregional lymph node metastasis [2,8]. However, there is no clear prognostic role for imaging modalities in these patients [2,5]. ^18^F-fluorodeoxyglucose (^18^F-FDG) positron emission computed tomography (PET/CT) is an imaging technique with a high accuracy in the detection of distant malignant melanoma soft-tissue and visceral metastases at initial staging or during follow-up [3,12,13]. Having said that, the well-known limitation of ^18^F-FDG PET/CT is its unfavourable tumour-to-background uptake ratio in the brain, making it a non-satisfactory modality for evaluating this region [3]. Instead, magnetic resonance imaging (MRI) is the modality of choice for the diagnosis of brain metastases [14].

Therefore, the ideal strategy for the whole-body assessment of metastatic melanoma at the time of diagnosis seems to be the combination of ^18^F-FDG PET/CT and brain MRI [13]. However, according to the ESMO guideline [15], brain MRI should be only applied to melanoma patients with a high risk for brain metastatic involvement. Also, the NCCN guideline recommends MRI only at the onset of neurological symptoms and when diagnosis would affect treatment decisions [14]. Thus, ^18^F-FDG PET/CT is the mainstay assessment tool for distant metastasis in melanoma patients, and consequently, the prognostic value of initial staging ^18^F-FDG PET/CT-derived parameters is important to determine, especially considering the brain [3].

The standardized uptake value (SUV) is the most widely used semi-quantitative parameter in ^18^F-FDG PET/CT, representing the degree of metabolic activity in a lesion [16]. As this value is heterogeneous within a tumour, the role of metabolic tumour volume (MTV) and total lesion glycolysis (TLG) as surrogates of tumour burden have been investigated and demonstrated to be a strong prognostic imaging marker in several oncologic conditions [2]. The prognostic predictive value of these metabolic parameters is increasingly under investigation in melanoma patients, and they were found to correlate with tumour aggressiveness and biological behaviour, being impactful on patient prognosis [3,13].

To the best of our knowledge, in terms of prognostication, the majority of previous studies have included small populations of advanced-stage melanoma patients. In addition, data regarding predictive value of baseline ^18^F-FDG PET/CT parameters for brain metastases in melanoma patients are scarce, especially when it comes to survival assessment. In this study, we aimed to investigate the prognostic value of baseline ^18^F-FDG PET/CT-derived parameters in predicting the occurrence of brain metastases in melanoma patients. 

## 2. Materials and Method

### 2.1. Study Population

A retrospective single-centre study was designed and received ethics committee and IRB approval. Searching our department’s medical records database, we reviewed 201 consecutive patients diagnosed with pathology-proven malignant melanoma between 2008 and 2021. Patients who underwent ^18^F-FDG PET/CT as a part of their initial staging were found to be eligible. Exclusion criteria included patients with a history of malignancy, unavailability/uncertainty in the initial stage of the disease, simultaneous diagnosis of infectious disease at the time of ^18^F-FDG PET/CT, presence of atypical tumours in their primary pathology, and having brain metastasis at the time of diagnosis, as well as those who developed brain metastases but revealed a second primary malignancy (e.g., lymphoma, breast cancer, or colon carcinoma) at the time of presentation or during follow-up.

Overall, 159 patients were enrolled. Patients’ initial stage assessment was based on histopathology, ^18^F-FDG PET/CT, and brain MRI within 3 months of diagnosis. Melanoma staging was determined according to the American Joint Committee on Cancer (AJCC; 8th edition) [17]. Clinical variables such as age, gender, location of the primary lesion, Breslow thickness, BRAF mutation, disease stage, years of follow-up, months to brain metastasis presentation, as well as treatment strategies, were also gathered. All patients underwent standard treatment based on their initial stage of the disease according to ESMO/NCCN guidelines. Additional immunotherapy, radiotherapy, or chemotherapy as adjuvant therapies were considered based on case-by-case decisions made by expert melanoma dermatologists.

### 2.2. ^18^F-FDG PET/CT Acquisition

All patients underwent our centre’s standardized protocol for in-line PET/CT imaging using one of two hybrid PET/CT systems: the Siemens 923/Biograph 64 mCT (Siemens Healthineers, Chicago, IL, USA) and the Philips Ingenuity TF/Gemini TF 16 (Philips Medical Systems, Andover, MA, USA). Both PET/CT scanners were accredited by EARL/EANM (EANM Research Ltd./European Association of Nuclear Medicine; Vienna, Austria). Acquisition and reconstruction parameters were harmonized between the scanners to minimize differences in SUVs (monthly cross-calibration of the instruments was performed in our centre). Prior to the scan, patients fasted for a minimum of 6 h. Just before tracer injection, serum blood glucose levels were documented to be below the threshold of 150 mg/dL. Otherwise, patients were rescheduled. Patients had to avoid substantial physical activity at least one day prior to the examination. Following mean uptake time (60 ± 5 min) after intravenous injection of an average dose of 4 MBq/kg ^18^F-FDG, a non-contrast-enhanced low-dose CT was performed for attenuation correction and precise anatomical location. CT scans were acquired using the two before-mentioned devices with acquisition characteristics of (a) Siemens Healthineers: Care Dose 4D; 100 kV, 59 mAs, 1.2 mm slice thickness, 1.5 pitch and (b) Philips Medical Systems: 100 kV, 33 mAs, 1.5 mm slice thickness and 0.8 pitch 0.8. Both scanners used the slice measuring 3 mm iterative algorithm (Siemens Healthineers: PSF + TOF and Philips Medical Systems: BLOB-OS-TF) for reconstruction purposes. The CT scan was followed by a 3 min per bed position, true whole-body PET acquisition [18].

### 2.3. Image Analysis

Areas of pathologic ^18^F-FDG uptake were assessed qualitatively and semi-quantitatively. All semi-quantitative analyses and delineations of tumoral lesions were performed by an expert on nuclear medicine and a molecular imaging specialist with knowledge of melanoma diagnosis. The semi-quantitative ^18^F-FDG PET-derived metabolic parameters of the primary tumour (non-resected cases), lymph nodes, and metastatic lesions were measured separately in each detected lesion. The extracted parameters included SUVmax, SUVmean of pathologic lesions including the prominent lesion with the highest uptake (pSUVmax and pSUVmean), maximal SUV corrected for the lean body mass (SULmax), SULpeak, pSULpeak, metabolic tumour volume (MTV) and total lesion glycolysis TLG (SUVmean multiplied by the corresponding MTV), as well as the number of tumoral lesions. Notably, MTV was measured by the percentage threshold-based semi-automatic edge detection method derived from the spherical volume of interest (3D VOI). Based on previous experiences, threshold cut-offs ranging from 40% to 60% have been used to provide the best visual contouring boundaries of the target lesion and to ensure no adjacent normal structures were included. Zero/near zero total MTV and total TLG was considered as a negative ^18^F-FDG PET/CT study.

### 2.4. Patient Follow-Up

All patients had serial follow-ups for diagnosing brain metastasis, including physical examination and imaging. Imaging studies were acquired every 6–12 months and/or when it was clinically indicated. All brain metastases were documented by brain MRI. Patient surveillance included a complete skin examination (plus a complete ophthalmic assessment in cases with ophthalmic melanoma) every 6–12 months for stages 0 and 1, as well as every 3–6 months for stage IIB and above in the first two years and every 6–12 months from 3 to 5 years. Brain MRI was considered every 3–12 months for the first 2 years and every 6–12 months from 3 to 5 years for stage IIB and above. For stages 0 to IIA, brain MRI was recommended when signs/symptoms were suspicious of metastatic disease [19]. All patients were followed up until they presented brain metastasis or were deceased without brain metastasis.

### 2.5. Statistical Analysis

Data were checked for consistency and normality. Continuous and categorical variables were reported as mean/median and relative frequencies (with percentage), respectively. Standard deviation (SD) and ranges were also reported if required. To compare groups, patients were divided into two main subgroups; patients who developed brain metastasis in follow-up and patients who did not. Noteworthily, in some of the scenarios mentioned in the results section, patients with non-cutaneous melanoma were excluded from the analyses to limit the study population to patients who only had cutaneous melanoma. Fisher’s exact test or Pearson’s chi-square test were used for cross-tabulation tables. The effects of ^18^F-FDG PET-derived parameters, as well as clinical/histopathological factors, were analysed on the cumulative incidences of brain metastases. We followed up with patients who did not present with brain metastasis until the day they were deceased. As death and brain metastasis are two competing events, competing risk regression models (Fine–Gray subdistribution proportional hazard model for competing risks) were used to analyse the effects of various covariates on the cumulative incidences of brain metastasis over time. To quantify the effects of the covariates, hazard ratios with 95% confidence intervals were computed. In all analyses, a two-tailed *p*-value of <0.05 was considered statistically significant. All statistical analyses in this report were performed using NCSS (NCSS 2022, NCSS, LLC. Kaysville, UT, USA), PASW 29 (IBM SPSS Statistics for Windows, Version 29, Armonk, NY, USA), and STATISTICA 13 (Hill, T. & Lewicki, P. Statistics: Methods and Applications, StatSoft, Tulsa, OK, USA) and the package ‘cmprsk’ [20] in R.

## 3. Results

### 3.1. Overall Characteristics and Clinical Factors

A total of 159 eligible melanoma patients (mean age: 70 ± 15; 55% male and 45% female) who underwent ^18^F-FDG PET/CT entered the study. Table 1 shows the detailed demographic characteristics.

All patients were between the ages of 31 and 97, and 75% of them were equal to or less than 81 years old. The majority of patients (85%) were in stages II (26%) and III (59%). Twenty patients (13%) had metastatic disease (stage IV) at the primary staging, including bone, liver, and lung metastases. The median follow-up time from the date of the primary tumour diagnosis or death without brain metastasis was 6 ± 3 years, with a range of one to 23 years. Age and gender did not show a significant correlation in the occurrence of brain metastases. The correlations of the studied clinical factors with brain metastasis development, as well as their HRs, are provided in Table 2. Among the variables, the initial M-stage and initial TNM stage were significantly correlated with brain metastasis development.

### 3.2. ^18^F-FDG PET/CT-Derived Parameters

Regarding the correlations of ^18^F-FDG PET/CT parameters with the occurrence of brain metastases, tumoral metabolic parameters with statistically significant differences are provided in Table 3 for all patients (n = 159) and in Table 4 after the exclusion of patients with non-cutaneous melanoma (n = 146). As shown in the tables, in both analyses, regional lymph node metastases’ SUVmax, SULmax, SUVmean, and SULpeak were significantly correlated with the outcome. Notably, all had a positive correlation with brain metastasis development. Similarly, prominent SULmax (pSULmax; only in the cutaneous + non-cutaneous cohort) and prominent SUVmean (pSUVmean; in both cutaneous +/− non-cutaneous cohorts) correlated with brain metastasis. Noteworthily, by limiting analyses to the patients who were in stages II and III (n = 135), regional lymph node metastasis SULmax and pSULmax retained their significance. However, there was no significant effect by limiting the analysis to only the BRAF-positive patients (n = 62).

### 3.3. Patient Outcome

During follow-up, 59 patients showed brain metastases. There were significant correlations between pSULmax (*p*-value 0.002 *; HR = 1.06; 95%CI: 1.02–1.10) and pSULpeak (*p*-value 0.014 *; HR = 1.06; 95%CI: 1.01–1.11) of the initial PET with brain metastasis (Figure 1). The higher the ^18^F-FDG PET/CT-derived parameter values, the more likely the patient was to develop brain metastases (Figure 2 and Figure 3).

pSULpeak was a significant semi-quantitative measurement considering patient prognostication in the multivariable assessment. Cumulative incidence curves for brain metastases based on pSULpeak as covariate (Figure 4) classified patients into two categories representing their risk levels: low to intermediate (≤15), and high (>15). It can be seen that patients with a high pSULpeak surge in terms of brain metastasis presentation.

## 4. Discussion

In melanoma patients, identification of those who are at higher risk of brain metastasis occurrence can be crucial as subsequent surveillance and even prophylactic treatment options can be provided. Early detection and treatment of asymptomatic brain metastatic lesions also minimize the treatment risks and achieve better outcomes [21]. Thus, in this study, we tried to evaluate potential prognostic factors in patients’ baseline assessment to predict brain metastasis development in melanoma, including histopathology and clinical factors, as well as ^18^F-FDG PET/CT findings.

Regarding histopathology and clinical factors, our study showed that there was no significant association between the primary tumour’s Breslow thickness and brain metastasis. This was in contrast with previous studies indicating tumour thickness as a clinical parameter for adverse prognostic factors [2,4,8]. Also, we did not find any significant relationship between sex and brain metastasis. Although sex is accepted as a prognostic factor for melanoma relapse and death, there is uncertainty as to whether the predominance of men with brain metastasis is due to behavioural factors or an inherent biological difference [22]. However, based on the current literature, melanoma is not a hormone-dependent malignancy, and sex was shown to not independently predict the pattern of metastasis and aggressiveness of the primary tumour, a fact supported by our findings [23,24].

Moreover, there was no significant correlation between BRAF mutation and the occurrence of brain metastasis in our study. This was also consistent with previous studies that showed no survival difference between BRAF mutation-positive or -negative tumours [8,25]. Furthermore, an immunohistochemistry study showed that BRAF mutation might not be involved in the brain metastatic cascade [26] and, in the literature, only a specific variant of BRAF (V600) was shown to have an association with the trend for shorter brain metastasis development [9,27]. However, the patients’ BRAF status in the research investigations is usually based on the primary tumour tissue samples. Nevertheless, in vivo tumour heterogeneity, intra-tumoral, and intra-patient BRAF mutation have been reported in the primary setting as well as recurrent and/or progressive diseases. Hence, it was proposed that for multiple primary melanomas or metastatic cases, multiple specimens are preferred over a single primary specimen for the BRAF mutation detection [28,29]. Thus, the evaluation of BRAF mutation status in, for example, ^18^F-FDG PET/CT-positive metastatic lymph nodes instead of the primary tumour, may have a potential value to be considered for further investigations.

Regarding ^18^F-FDG PET/CT findings, our results showed a significant relationship between the initial stage of the disease and brain metastasis. Additionally, regardless of the initial stage, the initial ^18^F-FDG PET-derived metabolic burden had predictive value in brain metastasis. Patients who had no obvious lesion on initial ^18^F-FDG PET/CT (zero/near zero total MTV and total TLG) had a high negative predictive value for brain metastasis (Table 3; HR = 0.43). To our knowledge, there is no previous study dedicated to predictive value and the relationship between ^18^F-FDG PET/CT and melanoma brain metastasis. However, previous studies indicated a high negative predictive value (97%) of ^18^F-FDG PET/CT for follow-up and detection of relapse in treated patients [29,30].

Furthermore, our results indicated that metastatic disease (M1) on ^18^F-FDG PET/CT, superior to the volumetric parameters and the number of detected lesions, was significantly correlated with brain metastasis (Table 2; HR = 2.92, 95%CI: 1.52–5.61). It was similarly shown that more than two metastatic organs were associated with shorter progression-free survival and had a strong prognostic factor for both progression-free survival and overall survival [2]. However, a few studies also showed MTV and TLG, as well as the number of metastatic lesions, could be used as surrogate markers for the prediction of recurrence and survival in melanoma patients [29,31,32], which seems contrary to our results as we did not observe a significant relationship between the total tumour burden in ^18^F-FDG PET/CT-positive patients and brain metastasis. However, melanoma has a stem cell-like phenotype and unique gene expression profiles, making it intratumorally heterogeneous. It can be disseminated from a relatively small primary tumour. Additionally, the precursor mutations driving brain metastases may differ from those promoting metastasis in other organs [33,34,35]. This may suggest that, for predicting brain metastasis in particular, relying on maximal/peak uptake values as well as the very presence of distant metastasis can be of more benefit than calculating tumoral volumes or counting the number of tumoral lesions throughout the body.

In this study, regional metastatic lymph nodes’ SULpeak and SULmax also had significant correlations with our groups (patients with brain metastasis development versus those who did not). These results were achieved regardless of the stage of the disease, BRAF status, age, and sex. Additionally, they retained their significance after the exclusion of patients with primary non-cutaneous disease. Thus, the characteristics of lymph nodes may have prognostic implications, especially considering that melanoma prognosis is closely tied to lymph node involvement [5]. The most common site (almost 75%) of metastases is regional lymph nodes, serving as an early indicator of distant involvement potential [12,36,37]. However, the metabolic information of lymph nodes did not add value to our final survival analysis, needing further investigations to support its value in brain metastasis predictions.

As a novel result, we found that pSULpeak was the best ^18^F-FDG PET/CT parameter in predicting brain metastasis. Thus, we aimed to identify trends and cut-off values in this regard. The higher the pSULpeak, the sooner the patient developed brain metastases. For example, if a patient had a high pSULpeak value of 19, then the risk of brain metastasis development was 80% after approximately 53 months, and with a pSULpeak value of 26, the patient had an 80% risk after about 39 months. Then, by stratifying patients into groups based on their pSULpeak on the primary ^18^F-FDG PET/CT, it was possible to clearly classify them into two risk groups: low risk (≤15), and high risk (>15).

We are aware of some limitations in this study. The major ones were, first, its retrospective nature, which could affect the results. Second, the single-centre design might have an impact on our findings, making its generalizability limited. However, we followed well-established guidelines for patient management to mitigate this issue as much as possible. Third, we included a relatively limited sample size. Although our population could be considered larger than most of the previous studies in this field, still larger cohorts are required to perform more accurate calculations (e.g., narrower CI for HRs). Additionally, the heterogeneity in the stages of the disease was a double-edged sword, and larger patient cohorts in each stage may provide more robust results.

## 5. Conclusions

In conclusion, our results may suggest that in addition to clinical and pathological characteristics of tumours, ^18^F-FDG PET/CT findings can be of value for prognostic risk stratification considering the occurrence of brain metastasis. A combination of metabolic information provided by baseline ^18^F-FDG PET/CT at initial staging, such as pSULpeak, may represent factors that can be used in predictive models. In summary, as the baseline pSULpeak increased, the likelihood of the patient developing brain metastases rose and occurred at an earlier time. Thus, our findings may help clinicians reach a better patient selection, identifying those who benefit more from regular MRI follow-up or additional prophylactic treatments to achieve more reliable brain surveillance in melanoma patients.

## Figures and Tables

**Figure 1 cancers-16-00127-f001:**
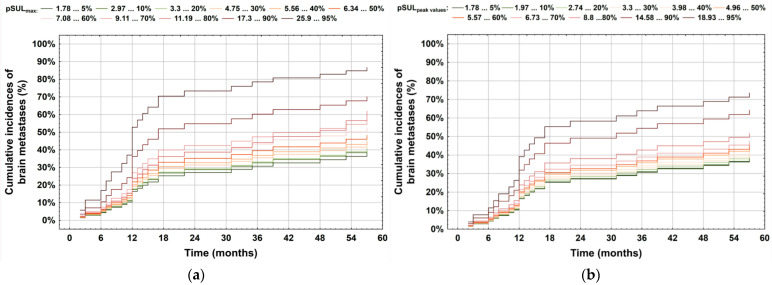
Correlation between occurrence of brain metastases and the impact of (**a**) pSULmax and (**b**) pSULpeak values. As depicted, the brain metastasis risk increased with higher values.

**Figure 2 cancers-16-00127-f002:**
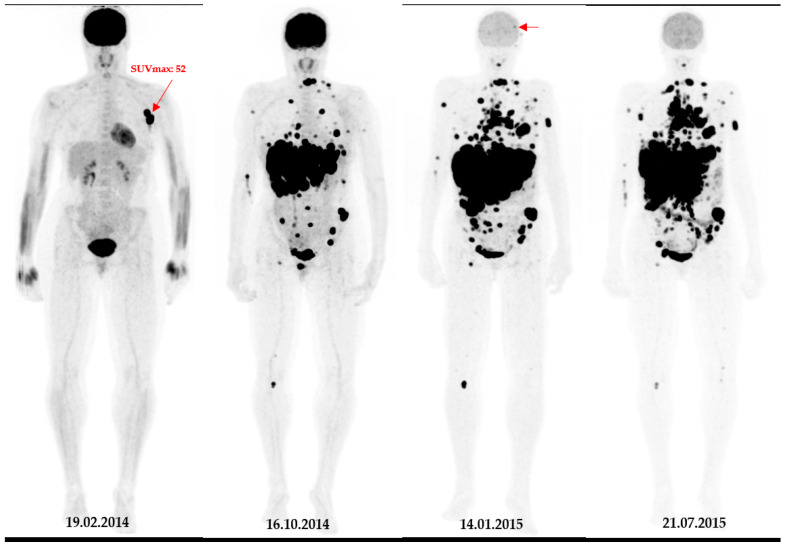
A 49-year-old patient with primary clinical stage III melanoma underwent ^18^F-FDG PET. The primary tumour Breslow thickness was 6.5 mm. On initial ^18^F-FDG PET images, this patient had a high SUVmax (=52), resulting in limited survival with the occurrence of brain metastasis in the first year of follow-up (horizontal arrow). Notably, during the course of the treatment, the patient underwent surgical removal of the primary tumour as well as lymph node dissection, followed by neoadjuvant radiotherapy and immunotherapy.

**Figure 3 cancers-16-00127-f003:**
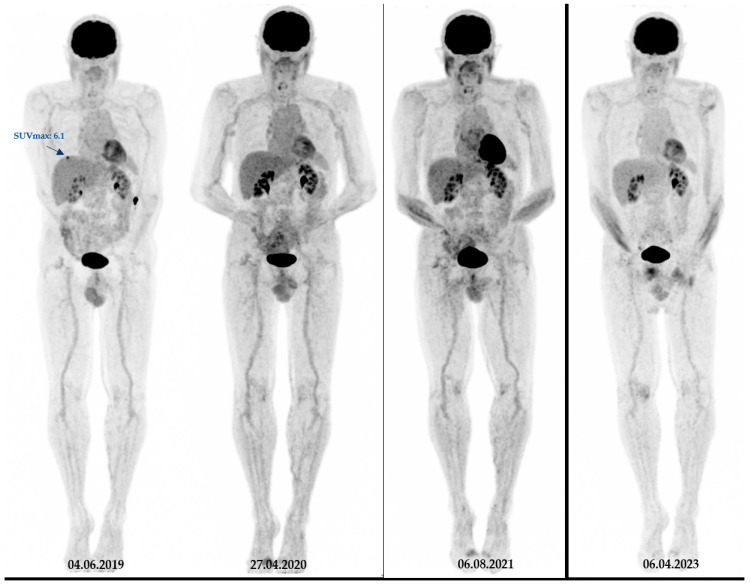
Another representative case from our studied cohort. A 73-year-old patient with melanoma (Breslow thickness = 3.5 mm) with initial stage IV disease. On baseline ^18^F-FDG PET images, this patient had a low SUVmax (=6) and was brain metastasis-free for at least four years of follow-up. Notably, during the course of the treatment, the patient underwent surgical removal of the primary tumour as well as lymph node dissection, followed by Interferon therapy and immunotherapy.

**Figure 4 cancers-16-00127-f004:**
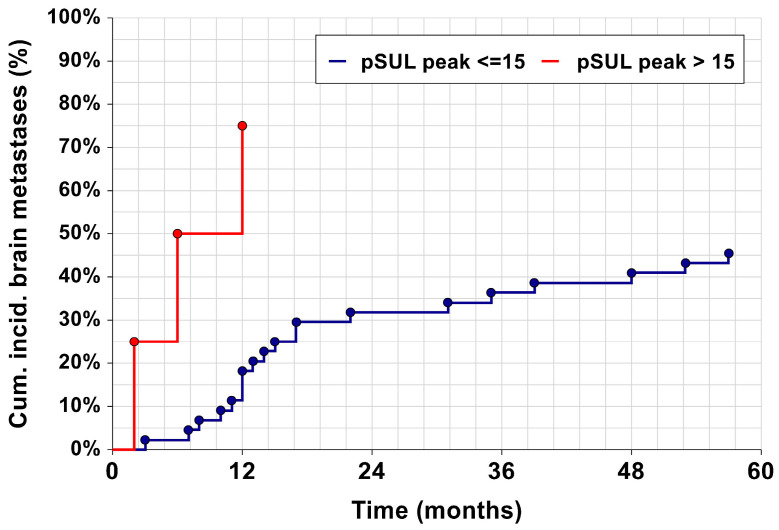
Kaplan–Meier curves, showing the cumulative incidence of brain metastases. The dark red line represents a pSULpeak of ≤15, and the blue line represents a pSULpeak > 15. Both curves are significantly different (*p* = 0.035).

**Table 1 cancers-16-00127-t001:** Detailed characteristics of the studied population (n = 159).

	Number (%)/Mean (SD)
Patients - Cumulative incidences of brain metastasis after 1, 5, and 10 years (%)	1 year: 10% (6.3–16.0%) 5 year: 30.8% (24.4–38.9%) 10 year: 35.2% (28.5–43.5%)
Age (years) at the time of baseline PET/CT	69.5 (±14.6); Range: 31–97
Sex - Female - Male	72 (45.3%) 87 (54.7%)
Location of the primary tumour - Skin - Other sites (uveal, mucosal, unknown)	146 (91.8%) 13 (8.2%)
BRAF mutation status - Positive - Negative	62 (39%) 97 (61%)
Immunotherapy history (before brain metastasis)	111 (69.8%)
Breslow thickness (mm)	3.49 (±3.25); Range: 0–14.5
Initial TNM stage - I - II - III - IV	4 (2.5%) 41 (25.8%) 94 (59.1%) 20 (12.6%)
Follow-up (years)	Median: 6.28; Range: 1–23

**Table 2 cancers-16-00127-t002:** Correlation of clinical variables and brain metastasis development.

	Hazard Ratio (95%CI)	*p*-Value
Age	1.00 (0.99–1.02)	0.70
Sex (reference: female)	1.02 (0.61–1.7)	0.93
BRAF positivity	1.31 (0.78–2.23)	0.31
Primary tumour’s Breslow thickness	1.08 (0.99–1.18)	0.081
Initial T stage (reference: T4)		
T1	1.0 (0.77–1.31)	1.0
T2	0.81 (0.55–1.17)	0.27
T3	0.93 (0.49–1.72)	0.80
Initial N stage (reference: N3)		
N0	0.82 (0.48–1.41)	0.48
N1	0.92 (0.45–1.85)	0.82
N2	0.69 (0.32–6.67)	0.63
Initial M stage (reference: M0)		
M1	2.92 (1.52–5.61)	<0.001 *
Initial TNM stage of disease (reference: stage IV)		
Stage I	0.8 (0.55–1.16)	0.66
Stage II	0.74 (0.51–1.07)	0.11
Stage III	0.36 (0.18–0.70)	0.003 *

* Statistically significant.

**Table 3 cancers-16-00127-t003:** Correlation of ^18^F-FDG PET/CT-derived parameters with brain metastasis development (n = 159).

Parameter	Hazard Ratio (95% CI)	*p*-Value
Regional lymph node metastasis SUVmax	1.02 (0.98–1.06)	0.44
Regional lymph node metastasis SULmax	1.07 (1.03–1.12)	<0.001 *
Regional lymph node metastasis SUVmean	1.04 (0.93–1.17)	0.51
Regional lymph node metastasis SULpeak	1.13 (0.98–1.30)	0.08
Regional lymph node metastasis metabolic tumour volume	1.00 (0.99–1.01)	0.69
Regional lymph node metastasis total lesion glycolysis	1.00 (1.00–1.00)	0.38
Number of lymph node metastases	1.02 (0.95–1.10)	0.63
Number of metastases other than lymph nodes	1.07 (0.97–1.18)	0.20
Patient’s prominent SULmax	1.08 (1.04–1.11)	<0.001 *
Patient’s prominent SUVmean	1.17 (1.06–1.30)	0.003 *
Patient’s prominent SULpeak	1.08 (1.04–1.13)	<0.001 *
Patient’s prominent metabolic tumour volume	1.00 (0.96–1.04)	0.99
Patient’s prominent lesion glycolysis	1.00 (0.99–1.00)	0.66

* Statistically significant based on competing risk regression models.

**Table 4 cancers-16-00127-t004:** Correlation of ^18^F-FDG PET/CT-derived semi-quantitative parameters with brain metastasis development in patients with cutaneous melanoma (n = 135). Only significant parameters are provided.

Parameter	Hazard Ratio (95% CI)	*p*-Value
Regional lymph node metastasis SULmax	1.11 (1.05–1.16)	<0.001 *
Patient’s prominent SULmax	1.05 (1.01–1.10)	0.028 *

* Statistically significant based on competing risk regression models.

## Data Availability

Data is accessible from the corresponding author upon request.

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
