# Peer review of "Prognostic Value of Baseline 18F-FDG PET/CT to Predict Brain Metastasis Development in Melanoma Patients"

_cancers, 2023, doi:10.3390/cancers16010127_

Round 1

Reviewer 1 Report

Comments and Suggestions for Authors

This is an interesting report about a retrospective study that aims at predicting the occurence of brain metastases in melanoma patients from FDG-PET findings at initial staging.

However, I have some major methodological points to make.

1) Type of data: It is not clear whether you use censored data which is the usually the case in this type of studies, or uncensored date. In any case, the contruction of the two groups (with/without brain mets) is only valid if you define a single follow-up time point at which you determine the absence/occurence of brain mets

2) Endpoint: You want to predict the occurence of brain metastases, but instead you use an endpoint (brain-metastasis free survival) where patients who die without brain mets are also counted as event. Such, the analysis could be driven by the patients dying from non-brain-related causes. This is probably not want you want to show, so you should add the  'freedom from brain metastasis' as endpoint where these patients are censored. Usually, the curves are then shown with the increasing incidence of BM pointing upwards.

3) It remains unclear how much the FDG-PET findings add to the more readily available clinical/histological findings. So I would suggest to use appropriate multivariate Cox Regression models that potentially show the independent prognostic value of the FDG-PET findings.

Minor points:

4) All tables are missing in the manuscript
5) It is not totally clear what exactly is meant with the total SUL max/ total SUV mean, is this from all lesions ?
6) Fig. 4: Overall survival or brain-mets-free-survival shown ? How many patients are in each group? Censored data ?
7) page 7, lines 249-256: these interpretation do not make sense; if the curve has a plateau, no more events are observed, so risk is zero

Comments on the Quality of English Language

Minor editing needed

Author Response

Reviewer 1

This is an interesting report about a retrospective study that aims at predicting the occurence of brain metastases in melanoma patients from FDG-PET findings at initial staging.

However, I have some major methodological points to make.

1) Type of data: It is not clear whether you use censored data which is the usually the case in this type of studies, or uncensored date. In any case, the contruction of the two groups (with/without brain metastasis) is only valid if you define a single follow-up time point at which you determine the absence/occurence of brain metastasis

Author reply: Thank you for raising this important point. In our previous analyses, we used Kaplan-Meier and Cox regressions to analyze brain metastasis-free survival, and we had censored data because patients who died without brain metastasis were censored (which was a mistake that we corrected later by using a competing risk regression model by Fine & Gray, please see below). We agree that a categorization based on censored data is not possible in this case. However, we followed your recommendation in 2): now we analyzed cumulative incidences for brain-metastasis using competing risk regression models. Using this model, we have two competing events: the occurrence of brain metastasis and death without brain metastasis. As all patients were followed up until they presented brain metastases or were deceased (without BM), we now have no censored data at all because exactly one event (BM or death without BM) occurred for each patient.

Nevertheless, we think it was not necessary to define two groups (with/without brain metastasis). Instead, we analyzed the effect of various parameters on the occurrence of brain metastasis. Due to these reasons, we have removed results for BM-.

2) Endpoint: You want to predict the occurence of brain metastases, but instead you use an endpoint (brain-metastasis free survival) where patients who die without brain metastasis are also counted as event. Such, the analysis could be driven by the patients dying from non-brain-related causes. This is probably not want you want to show, so you should add the  'freedom from brain metastasis' as endpoint where these patients are censored. Usually, the curves are then shown with the increasing incidence of BM pointing upwards.

Author reply: Thank you for raising this important issue. We discussed your very valuable arguments with a statistician, and we had to realize that Cox regression and Kaplan-Meier methods are not useful for analyzing the occurrence of brain metastases. We had to learn, and we fully agree that these methods did not take those patients properly into account who died without having brain metastasis. We also agree that the previous analyses might have been driven by patients dying from non-brain-related causes.

Thus, we followed your advice and discussed which statistical model properly takes those patients into account who died without brain metastasis without counting them as an event. We decided to apply ‘competing risk regression models’ and reevaluated all results. These models work by using two competing events: ‘occurrence of brain metastases’ and ‘death without brain metastases’.

Additionally, we present the results using increasing cumulative incidence curves for the occurrence of brain metastases, as you recommended.

We also compared the results in Fig.1b based on the previous analyses (BMFS and Cox-regression) and compared it with the new model results (occurrence of BM and Fine Gray model) and found that the cumulative incidences estimate that the risks for BM are smaller – in accordance with your remark that the results may be driven by patients dying from non-brain-related causes.

To the best of the authors' knowledge, the computation of median time to occurrence of brain metastasis is not feasible when using competing risk regression models. We hope you will find this analysis suitable.

Authors action: The competing risk regression model (‘proportional hazards model for the subdistribution of a competing risk by Fine & Gray’) was used to reanalyze all results, including Tab.2, Tab. 3, Tab.4, Fig.1a,b and Fig.4 as well as all other results for the occurrence of brain metastasis. 

We thank you for raising this important issue!

3) It remains unclear how much the FDG-PET findings add to the more readily available clinical/histological findings. So I would suggest to use appropriate multivariate Cox Regression models that potentially show the independent prognostic value of the FDG-PET findings.

A: Thanks for your valuable comment. Our purpose in this study was not to provide a comprehensive prediction model for brain metastasis prognostication. Instead, we wanted to evaluate if there is any potential for baseline FDG PET imaging to predict vulnerable individuals (as the first step in the literature). Thus, we did not include clinical/histopathological data in our multivariate analysis as we already know their value, and as calculated in our study, for instance, M1 had a hazard ratio of 3.2, which obviously would be the single best predictive parameter in the multivariate analysis. Hence, talking about PET parameters alongside M1 status, for example, would push our aims out of the spotlight as PET-derived factors could not be found as strongly prognostic as M1 status. Thus, we only focused on PET parameters on their own to investigate their potential value and guide future studies and not to develop a prediction model. Hope you accept our explanation in this regard.

Minor points:

4) All tables are missing in the manuscript

A: Thanks. We added all the tables.

5) It is not totally clear what exactly is meant with the total SUL max/ total SUV mean, is this from all lesions ?

A: Thank you for your comment. The total SUVmax and SUVmean refer to the values of the most-avid tumoral lesion in the body. However, in order to avoid any misinterpretation we have changed the word “total” to “prominent” (pSUVmax, pSUVmean and pSULpeak).

6) Fig. 4: Overall survival or brain-metastasis-free-survival shown ? How many patients are in each group? Censored data ?

Author reply: Fig.4 illustrated brain-metastasis-free-survival, there were 5 patients in the SUL-peak <2, 39 patients (SUL-peak 2-14.6) and 4 patients (SUL-peak > 14.6). As brain-metastasis-free survival was shown, data were censored.

After a complete reanalysis of all data following your suggestion, Figure 4 was reevaluated by using a competing risk model, and we found a significant difference. However, in the subgroup SUL-peak < 2, only a very small number of patients were found and – after discussion - we used only one cutoff, which is 15. Again, we found a significant difference (p=0.035*).

Author action: Data were reanalyzed, and Fig. 4 presents cumulative incidences of brain metastases for PET-positive patients with a total SUL peak <=15 and >15. Both curves were found to be significant (p=0.035*).

7) page 7, lines 249-256: these interpretation do not make sense; if the curve has a plateau, no more events are observed, so risk is zero

Author reply: Please note that the curve with the plateau occurred in the group with SUL-peak < 2, where the risk of developing at a later time seems to be low. Additionally, the sample size in this group was very low. As pointed out in 6), we applied the cut-off of 15, and here we see that the curve (total SUL peak <=15) is slightly increasing over time (especially after 18 months).

Reviewer 2 Report

Comments and Suggestions for Authors

This was a clear and well-presented study with a conclusion well evidenced by the results. I have two small questions for the authors:

1. Why was not mean SUV corrected for the lean body mass "SULmean" tested as a paramater for correlation?

2. Would it make sense to use a parameter like "total SUV", calculated by SUVmean times the tumor volume? It would seem to be the closest measure of total number of tumor cells in the body? If it would not make sense, why not?

Author Response

Reviewer 2:

This was a clear and well-presented study with a conclusion well evidenced by the results. I have two small questions for the authors:

  1. Why was not mean SUV corrected for the lean body mass "SULmean" tested as a paramater for correlation?

A: Thank you for your comment. We have tried to assess the predictive value of the most relevant semi-quantitative PET parameters that are commonly used in routine clinical practice, such as SUVmax and SUVmean. However, as PERCIST and recently introduced modified criteria for treatment response assessment in melanoma patients are relying on SULpeak, we have included SULpeak and SULmax to comply with PERCIST criteria and follow a standard methodological approach for semi-quantitative assessment in FDG-PET/CT imaging of melanoma patients. We agree with you that SULmean may also provide some additional information; however, we think that adding more parameters may affect the readability of the article and seems not to change the final message of this study.

  1. Would it make sense to use a parameter like "total SUV", calculated by SUVmean times the tumor volume? It would seem to be the closest measure of total number of tumor cells in the body? If it would not make sense, why not?

A: Thanks for this interesting point. The total SUVmax and SUVmean refer to the values of the most-avid tumoral lesion in the body. However, in order to avoid any misinterpretation, we have changed the word “total” to “prominent” (pSUVmax and pSUVmean).

Regarding your suggestion, we have followed the known definitions and included TLG, which is total metabolic volume multiplied by SUVmean. We believe this is equivalent to what you suggested.

Round 2

Reviewer 1 Report

Comments and Suggestions for Authors

Thank you for this comprehensive re-analysis, the paper has now clearly improved.